# High-Throughput Identification of Putative Antimicrobial Peptides from Multi-Omics Data of the Lined Seahorse (*Hippocampus erectus*)

**DOI:** 10.3390/md18010030

**Published:** 2019-12-29

**Authors:** Xiyang Chen, Yunhai Yi, Xinxin You, Jie Liu, Qiong Shi

**Affiliations:** 1BGI Education Center, University of Chinese Academy of Sciences, Shenzhen 518083, China; chenxiyang@genomics.cn (X.C.); yiyunhai@genomics.cn (Y.Y.); youxinxin@genomics.cn (X.Y.); 2Shenzhen Key Lab of Marine Genomics, Guangdong Provincial Key Lab of Molecular Breeding in Marine Economic Animals, BGI Academy of Marine Sciences, BGI Marine, BGI, Shenzhen 518083, China; 3BGI Genomics, BGI-Shenzhen, Shenzhen 518083, China; liujie8@genomics.cn; 4Laboratory of Aquatic Genomics, College of Life Sciences and Oceanography, Shenzhen University, Shenzhen 518060, China

**Keywords:** lined seahorse, antimicrobial peptide (AMP), genome, transcriptome, proteome, sexual specificity

## Abstract

Lined seahorse (*Hippocampus erectus*), the most widely cultivated seahorse in China, has been in short supply because of its important medicinal value; meanwhile, unnatural deaths caused by various diseases (especially enteritis) have limited their practical large-scale aquaculture. Antimicrobial peptides (AMPs), as the best alternative to antibiotics, have been extensively applied in agricultural practices. In this study, we identified 290 putative AMP sequences from our previously published genome and transcriptome data of the lined seahorse. Among them, 267 are novel, and 118 were validated by our proteome data generated in the present study. It seems that there is a tissue preference in the distribution of AMP/AMP precursor transcripts, such as lectins in the male pouch. In addition, their transcription levels usually varied during development. Interestingly, the representative lectins kept extremely high levels at the pre-pregnancy stage while at relatively lower levels at other stages. Especially *Lectin25*, with the highest transcription levels and significant developmental changes, has been reported to be involved in seahorse and human pregnancy. The comparison of transcriptome data between one-day and three-month juveniles indicated that *Hemoglobin2* (*Hemo2*) was significantly upregulated in the body, haslet, and brain. Our proteome data of female and male individuals revealed three putative AMP precursors with sexual specificity, including two male-biased cyclin-dependent kinases (CDK-like16 and CDK-like23) and one female-biased bovine pancreatic trypsin inhibitor 2 (BPTI2). In conclusion, our present high-throughput identification of putative AMP sequences from multi-omics (including genomics, transcriptomics, and proteomics) data provides an overview of AMPs in the popular lined seahorse, which lays a solid foundation for further development of AMP-based fish food additives and human drugs.

## 1. Introduction

Seahorses, belonging to Syngnathidae, are well known for their special shape, male pregnancy, and pharmaceutical values [1]. Their shapes are usually characterized by an elongated snout, a body covered with hard bones but without scales, and a shortage of pelvic and caudal fins. Their reproductive behaviors are also unique to male pregnancy. In addition, seahorses have been a precious resource for traditional Chinese medicine [2,3]. According to many old records, at least five seahorse species had been applied in the production of these medicines; especially, the lined seahorse (*Hippocampus erectus*), introduced mainly from America, has been the most widely cultivated species in China. Recent studies have shown that extracts and isolates from seahorses have a variety of pharmaceutical activities, including antioxidant [4], immune-enhancing [5], sex hormone-like [6], anti-tumor [7], and anti-inflammatory [8] functions.

Seahorses have been in great demand due to their medicinal values, although they are mainly supplied by field fishing. Nevertheless, large-scale aquaculture of seahorses has been seriously limited by fatal diseases, which are usually caused by exogenous pathogens, such as vibrio [9,10,11] and ciliate [12]. Enteritis is a serious disease restricting the practical aquaculture of the lined seahorse with strong infectiousness and high fatality rates [11,13], especially occurring most frequently among juveniles during a short period from the newborn to approximately one month after birth. The popular antibiotics for fish aquaculture practices, however, often cause drug resistance [14], and residual drug issues. Thus, it is urgent to find a safer antibacterial alternative to improve the quality and quantity of cultivated seahorse products and sustainable development of these fish species.

Antimicrobial peptides (AMPs), a group of amphiphilic and cationic short peptides, are in the first line of defense for the host against infectious pathogens with extensive existence in organisms. Currently, the public database of antimicrobial peptides (APD3, http://aps.unmc.edu/AP/main.php) contains 3,138 AMPs from six kingdoms [15]. Most AMPs either have their own genes or come from the hydrolysis of other immune-related proteins. There is growing evidence for derivation of AMPs from the hydrolysis of some large proteins, like histone, lectin, thrombin, hemoglobin, and so on. For example, Buforin I, a histone H2A-derived AMP from the Asian toad (*Bufo gargarizans*), was proved to have antimicrobial activity [16]. The most known antibacterial mechanisms for these AMPs include cell membrane osmosis and internalization [17]; in the former, AMPs form transmembrane pore channels and damages to the cell membrane, while AMPs in the latter enter the bacterial cells directly and act on specific intracellular targets to inhibit the synthesis of nucleic acids and proteins. Antimicrobial activities of some AMPs are related to their structures. A review [18] already investigated and discussed major classes of AMPs in fishes; among them, β-defensins and hepcidins could give rise to a conserved β-sheet topology with several disulfide bridges [19,20]. Recently, some of these AMPs have been extensively studied and widely used in the fields of pharmacy, health care products, and agriculture practices [21,22,23]. As one of the best alternatives for antibiotics, AMPs have broad application prospects, especially in world-wide freshwater and marine aquaculture [23,24].

Our previous study [25] identified 507 AMPs/AMP precursors from the genome and transcriptome data of two amphibious mudskippers, *Boleophthalmus pectinirostris* (BP), and *Periophthalmus magnuspinnatus* (PM); among them, 23 AMPs/AMP precursors were screened out and synthesized for antibacterial experiments; hemoglobin β1- and amylin-derived AMPs were further proved to effectively inhibit the growth of *Micrococcus luteus*, indicating that relevant research is offering a theoretical basis for disease prevention in fishes. In the present study, we aim to identify putative AMP sequences from our previously published genome and transcriptome data of the lined seahorse in a high-throughput way, and then to check their transcriptional changes in various tissues during development. Production in the lined seahorse of some potential AMPs/AMP precursors is validated at the protein level based on our newly generated proteome data in the present study. Furthermore, differential gene expression between male and female individuals based on the proteome data was also analyzed for potential sexual variances and specificity.

## 2. Results

### 2.1. Summary of Achieved Data from Our Previously Published Genome and Transcriptomes

We collected 2,927 AMP amino acid sequences from the APD3 database (Appendix A), which was used as a local database of query sequences. Genome sequences and 19 transcriptome datasets of the lined seahorse from our previous reports [1,26] were employed as the seed database. The 19 transcriptome datasets cover serval tissues at different developmental stages, with each sample from a single lined seahorse (Table 1 and Table 2). After running BLAST and in-house script to search, filter and integrate our results, we obtained 256, 186 and 267 putative AMP sequences from the annotated gene set (i.e., the coding sequence of each predicted gene [26]), assembled genome scaffolds and transcriptome sequences, respectively (Appendix A). After removal of redundant sequences, we finally determined a total of 290 putative AMPs and AMP precursors in the lined seahorse (Appendix A). After removal of 18 sequences identified from genome scaffolds (for their long lengths), we listed full-length coding sequences of other 272 AMPs/AMP precursors with yellow marks of the aligned regions in Appendix A.

Statistics of these identified putative AMP/AMP precursor numbers from the 19 transcriptome datasets are summarized in Table 1 and Table 2. According to the annotations in the public APD3 database, we classified these putative AMPs/AMP precursors into 24 groups, including thrombin, histone, lectin, defensin, cyclin-dependent protein kinases (*CDK-like*) [27], antiproteinase (*antipro*) [28], bovine pancreatic trypsin inhibitor (*BPTI*) [29], phospholipase A2 (*PLA2*) [30], liver-expressed antimicrobial peptide 2 (*LEAP-2*) [31], Beta2-microglobulin (*β2m*) [32], enhancers of rudimentary (*ERH*) [33], glyceraldehyde 3-phosphate dehydrogenase (*GAPDH*) [34], and so on (see more details in Figure 1).

Among the 290 putative AMP sequences, 267 are novel (with at least one residue variance from the corresponding query; Appendix A), and only 23 are identical to the reported query sequences, such as those from six putative ubiquitins (Ubiq, 74-aa peptides originally isolated from the gill of Pacific oyster, *Crassostrea gigas* [35]) and 17 putative histone-derived AMPs (e.g., histone H4-derived AMPs isolated from American cupped oysters *Crassostrea virginica* [36], H3-derived AMPs isolated from calf thymus *Bos Taurus* [37], and H2A-derived AMPs isolated from hemocytes of Pacific white shrimp (*Litopenaeus vannamei*) [38]). As shown in Figure 1, thrombin (*Throm*) is the largest group of AMP precursors, with the mapping of 62 thrombin-derived C-terminal peptides (TCPs). In fact, TCPs usually aggregate with lipopolysaccharide (LPS) and Gram-negative bacteria to form an amorphous amyloid-like substance, contributing to the clearance of those aggregates by phagocytosis [39]. Recently, a high structure resolution of TCPs revealed the vital peptide regions for interaction with LPS and CD14, providing a detailed molecular explanation for the practical antimicrobial activities of TCPs [40].

The histone family, the second major group with the mapping of known AMPs in the present study, consists of histones *H2A*, *H4*, *H3*, and *H2B*. Lectins, also a major group among the putative AMP precursors with a reported role in male pregnancy [41], consist of 23 RegIIIgamma (a secreted C-type lectin from the intestine of the mouse) and nine RegIIIalpha (regenerating islet-derived protein 3-alpha from human) [42]. Others, including ubiquitin, *CDK-like*, *BPTI*, and chemokine (*Chem*), also accounted for a large proportion separately.

We performed the Kyoto Encyclopedia of Genes and Genomes (KEGG) [43] clustering analysis on the 272 newly predicted AMP sequences, including 256 from the genome gene set [26] and 16 from our reported transcriptomes [1]. These AMP sequences were searched against the KEGG database with BLAST. They were finally clustered into 193 KEGG pathways. Representative classes include “cancers: an overview”, “immune diseases”, “substance dependence”, and “immune system” (see Figure 2), indicating that most of the putative AMPs/AMP precursors are potentially involved in immune and disease resistance. In addition, we noticed that 27 and 24 putative AMP/AMP precursor genes were clustered into “infectious diseases: bacterial” and “infectious diseases: viral” terms, respectively, which may be relevant to antimicrobial defense.

### 2.2. Representative AMPs/AMP Precursors with High Transcription Levels in the Published Transcriptome Datasets

A total of 19 transcriptomes from several tissues of the lined seahorse were assembled from our previously reported raw data [1]. Finally, we obtained 267 putative AMP sequences by searching against these datasets (Appendix A). Corresponding transcription values (fragments per kilobase of transcript per million mapped reads, FPKM) were calculated for the quantification of mRNA levels. Top 20 putative AMPs/AMP precursors in each transcriptome were picked out (Appendix A) for comparison.

Interestingly, we observed that the transcription levels of some lectins were usually extremely high among these identified putative AMP/AMP precursor transcripts. Three with FPKM values over 18,000 are all lectins in the brood pouch. Especially, the FPKM values of *Lectin25* were the highest in the pouch at both pre-pregnancy (58,060.70) and pregnancy (24,092.77) stages, followed by *Lectin11* with an FPKM value of 21,205.30 and *Lectin20* with an FPKM value of 20,844.54 in the pouch at the pre-pregnancy stage. *Lectin20* also highly transcribed in the pouch at the rudimentary stage, with an FPKM value of 18,732.16.

Statistics of these Top 20 putative AMPs/AMP precursors revealed that histones and ubiquitins were highly transcribed in embryos; ubiquitins and neuropeptides (*Neuropep*) were the major classes in the brain; in the pouch, however, lectins and ubiquitins were with high transcription levels; in the testis, ubiquitins and histones accounted for the major part. It seems that ubiquitins were widely distributed in various tissues.

### 2.3. Transcriptional Changes of the Predicted AMPs/AMP Precursors: Based on Our Previously Published Transcriptome Datasets

Those Top 20 putative AMPs/AMP precursors with low transcription levels and insignificant changes (less than two folds) in the testis were removed, and the remaining data were displayed in Figure 3. In embryos, *Lectin20*, *Lectin23*, *Lectin21*, *Antipro2*, and *Antipro3* were upregulated, while *CDK-like12* and *CDK-like21* were downregulated during the period from 1 to 10 days post fertilization (ptf1-10); *histone42*, *histone4*, *histone3*, and *histone10* were highly transcribed in ptf1 and low transcribed in ptf3 and ptf10 (Figure 3a). In the brain, *Hemo4* (hemoglobin 4) is the most downregulated putative AMP precursor throughout the period from rudimentary to pregnant stages with very high transcription levels in the juvenile, rudimentary, and post-pregnancy stages. In addition, the synthesized peptide of hemoglobin β1, identified by the present high-throughput method, has been proven to have high inhibitive activities on *Micrococcus luteus* [25]. The peptide of hemoglobin β1 was downloaded to search against the gene set of the lined seahorse [26] with BLAST (an e-value of 1 × 10^−5^), and surprisedly we found that it matched to Hemo4 with an identity of 68.70% in the protein sequence. The developmental change trend of *Hemo2* was similar to *Hemo4* with a low transcription and change rangeability. *Transfe1* and *Ubiq22* were upregulated and then downregulated, and their transcription levels peaked at the pre-pregnancy and pregnancy stages, respectively, while *Throm63* maintained high transcription levels with a modest change rangeability (Figure 3b).

In the testis, *Throm64*, varying significantly, was downregulated until the pregnancy stage and then upregulated. *Ubiq16* was downregulated and then maintained a relatively stable level after a high transcription level at the pre-pregnancy stage. The transcription levels of *Hemo4*, *Hemo2*, *Ubiq22*, *Transfe1*, *Antipro3*, *Antipro2* and *β2m1* reached the highest at the pregnancy stage with a similar change tendency. Among them, *Antipro3* and *Antipro2* were lowly transcribed at other stages (Figure 3c). Furthermore, in the pouch, *Lectin25*, *Lectin11*, and *Lectin20* were upregulated and then downregulated, while their transcription levels at the pre-pregnancy stage reached the highest. In addition, the transcription levels of *Lectin25* were always extremely high; *Transfe1* was highly transcribed at the pregnancy stage but with low levels at other stages, and the transcription levels of *Hemo4* kept going up during development (Figure 3d).

Representative putative AMPs or AMP precursors with high transcription levels or significant changes (at least two folds) in each tissue were picked out. Their change patterns at different developmental stages were marked in Figure 4. We speculate that those putative AMPs and AMP precursors with high transcription levels or significant fold changes at a certain stage may play certain vital roles, which are worthy of in-depth investigations into their disease resistance for aquaculture practices.

According to previous studies, juvenile seahorses within one month after birth are more likely to be susceptible to exogenous bacterial infection, and enteritis frequently occurs in this period. Comparisons of transcriptional changes between 1-day and 3-month juveniles were therefore performed. We noted that the sequenced juvenile_body [1] was the mixture of pouch and trunk. Interestingly, we found that, compared with the one-day juvenile stage, *Lectin25*, *Hemo2*, *BPTI10*, and *Chem10* were upregulated significantly in the body (Figure 5a). In the haslet, *Antipro2*, *Transfe1*, *BPTI4*, *Throm15*, *Hemo2*, and *Antipro3* were upregulated significantly (Figure 5b). Furthermore, *Hemo2*, *Neuropep8*, *Throm63*, *Neuropep7*, *Ubiq2*, and *Transfe1* were upregulated significantly in the brain (Figure 5c). Notably, *Hemo2* was upregulated significantly in all examined tissues. *Transfe1* was upregulated in both haslet and brain, and it was predicted as eukaryotic translation initiation factor 4 gamma 1.

Since these significantly upregulated AMPs or AMP precursors may be involved in the process of antimicrobial defense during the first trimester, they are expected to benefit for improvement of survival rates of juvenile seahorses in practical aquaculture.

### 2.4. Validation of the Newly Generated Proteome Data

In this study, proteome sequencing of six male and six female lined seahorses was achieved (see more details in Section 4.3). Statistics of total spectra, sequenced peptides, and predicted proteins are summarized in Appendix A. MS.F.20 and MS.M.20 were detected by the routine mass spectrometry (MS) analysis for female (F) and male (M) individuals, respectively; their peptides were separated into 20 fractions before MS acquisition. MS.F.1 and MS.M.1 were similar for females and males, respectively, but without fractionation. Moreover, DIA_F and DIA_M covered identification and quantification of proteins by Data Independent Acquisition (DIA) for female and male seahorses, respectively (see more details in Section 4.3.2). After the integration of all the achieved results and removal of redundant data, we finally obtained a total of 6,340 proteins from the examined samples of the lined seahorse. Among them, 6055 proteins were determined for the female and 4,849 proteins for the male seahorses. We also mapped these proteome datasets to the predicted putative AMPs/AMP precursors, with confirmations of 118 (in total), 112 (female), and 94 (male) AMP sequences, respectively. Details of the total 118 AMPs/AMP precursors were listed in Appendix A.

To explore the sexual differences of putative AMP or AMP precursors, we compared them between male and female seahorses based on the proteome data. Interestingly, we found that 24 putative AMPs or AMP precursors were only detected in the female, and six AMPs or AMP precursors were only available in the male seahorses. Their transcription levels as well, as aligned regional sequences, are shown in Appendix A. However, we confirmed that several putative AMP/AMP precursor genes were also detectable in transcriptome data of both genders. Consequently, after validation with the transcriptome data, we determined three sexually biased putative AMP precursors, including two male-biased (CDK-like16 and CDK-like23) and one female-biased (BPTI2; Appendix A).

## 3. Discussion

### 3.1. Variances and Consistences of Putative AMPs/AMP Precursors from the Multi-Omics Datasets

We performed high-throughput identification of AMP sequences in the lined seahorse based on multi-omics data, including genome, transcriptome, and proteome datasets. The genome provides an overview of the potential AMPs/AMP precursors, and the transcriptomes provide transcriptional details of these genes; however, the proteome reveals the bioactive peptides (or derived) from these genes. As we discussed before [25], multi-omics data supply more chances to identify more AMPs/AMP precursors and to verify them from different views in gene expression. Especially, the proteome data provide strong evidence for verification of these AMP sequences. In the lined seahorse, we identified 118 putative AMP or AMP precursors based on the proteome datasets, which is different from the total number of genes or transcripts, although they are overlapped among the three resources of data (see more details in Figure 6).

The multi-omics data provide mutual verification and support; however, the quality of sequencing and assembly, as well as different analysis technologies, may lead to indeterminacy in the identified gene number. To confirm that the 16 novel predicted transcripts based on transcriptome data (Figure 6) are inconsistent with the genome-annotated gene set, we employed BLAST (an e-value of 1 × 10^−5^) to search against the gene set sequences and observed that these transcripts were not identical to any genes in the annotated gene set [26].

Transcriptome data provide clues for transcriptional changes of AMPs or AMP precursors at several developmental stages or in different tissues, and the proteome data validate proteins for potential production of AMPs. However, analyses based on multi-omics datasets also have limitations in the present study because many AMP sequences are from AMP precursors, although hydrolysis of these large proteins (such as histone, lectin, and thrombin) may not have always occurred. Hence, transcription analysis based on the transcriptome data can’t accurately reflect changes of AMPs, and validation based on only proteome data can hardly discriminate the presence of AMPs.

### 3.2. Species Difference in AMPs/AMP Precursors: Using the Lined Seahorse and Amphibious Mudskippers as the Representatives

AMP studies based on fish genomes and transcriptomes are rare, whereas identification of AMPs or AMP precursors from mudskippers, a group of representative amphibious fishes, were reported by us previously [25]. The comparison between putative AMP sequences from mudskippers and the lined seahorse demonstrated that these teleost fishes have similar putative AMP/AMP precursor numbers (BP, 257; PM, 250; the lined seahorse, 290), although the putative AMPs/AMP precursors identified from the lined seahorse transcriptome and proteome data are approximately twice as many as those from mudskipper transcriptomes.

Based on the detailed results, however, we found that putative AMP groups in the lined seahorse lack piscidin, hepcidin, lysozyme, RNase, misgurin [44], SK84, and lipid transfer protein (LTP). Interestingly, thrombin was the biggest group in both mudskippers and the lined seahorse, indicating their vital roles in fishes. The second biggest group in the lined seahorse, histone, was approximately twice as many as those in BP and three times more than those in PM (BP, 21; PM, 10; the lined seahorse, 52). Previous works have proven that histones have broad-spectrum activities against fish pathogens [45]. Our finding, therefore, suggests that histones may play important roles in the lined seahorse for defending against microorganisms. In general, the major groups of putative AMPs in mudskippers and lined seahorse are similar, but their numbers are a little bit different.

### 3.3. Representative AMPs/AMP Precursors: Tissue Distribution and Developmental Changes

#### 3.3.1. *Lectin25*

There exist remarkable differences for many representative AMPs/AMP precursors in tissue distribution and their transcription levels during development. In the pouch where male seahorses nourish embryos, lectins were usually the most abundant AMP precursors with high transcription levels. Among them, *Lectin25* showed obvious developmental changes with extremely high transcription levels, especially at the pre-pregnancy stage (Figure 7a). Based on the proteome datasets, we observed that the Lectin25 was detectable in the MS data but not in the DIA data. Gene structure prediction indicated that the *Lectin25* has six exons, and the AMP matching region started from the second exon and ended in the fourth exon (172–390th bp; see detailed protein sequence in Figure 7b). *Lectin25* was predicted to be “Galactose-specific lectin nattectin” by the Swiss-Prot annotation. In fact, nattectin belongs to the family of C-type lectins and was firstly isolated from the venom of a Brazilian venomous fish (*Thalassophryne nattereri*) [46]. Nattectin-like protein, a C-type lectin from Qihe crucian carp (*Carassius auratus*), was proven to have an agglutinating activity against bacteria [47,48] and play an important role in antimicrobial defense, also suggesting that this lectin may have antimicrobial activities.

According to previous work [41], *hc*CTL III, one of C-type lectins (CTLs) in the tiger tail seahorse (*Hippocampus comes*), was secreted into the pouch during early pregnancy; bacteriostatic tests showed that *hc*CTL III inhibited the growth of *Escherichia coli*, demonstrating its antibacterial activity. Gene set of the lined seahorse were searched against by BLAST (e-value = 1 × 10^−5^) with protein sequences of *hc*CTL I, II, and III as the queries, and those hits with the highest identity for each *hc*CTL sequence were picked out. Interestingly, we found that *hc*CTL III mapped to Lectin25 with an identity of 80.2% (Figure 8). It seems that *Lectin25* may play a significant role in embryo antimicrobial protection during male pregnancy [41,49]. Interestingly, lectins have been reported to be associated with human pregnancy. For example, a previous work [50] has proven that deficiency of mannan-binding lectin (MBL) was related to recurrent miscarriage, and MBL concentration significantly increased during pregnancy. Another work [51] found that codon 54 polymorphism of the *MBL* gene may lead to pregnancy-related problems.

*Lectin11* was also highly transcribed at the pre-pregnancy stage in the pouch with an FPKM value of 21,205.30. Interestingly, we found that both *Lectin25* and *Lectin11* located in the same scaffold45 with adjacent localizations. In summary, it seems that lectins are a vital resource of AMPs during seahorse and human pregnancy. We hence propose further studies on these genes to make a great contribution to at least the seahorse breeding and aquaculture.

#### 3.3.2. *Hemo2*

A comparison between 1-day and 3-month juveniles showed that *Hemo2* was significantly upregulated in the seahorse body, haslet, and the brain (Figure 5). Hemo2 was also detectable in the proteome data, with the relative quantitative values of 2,814,388 and 3,167,032 in female and male lined seahorse, respectively.

Hemoglobin-derived AMPs were proven to have broad-spectrum activities against pathogens in many previous works. For example, from specimens of human endometrium mucus, they were reported to present strong inhibitive activities against *E. coli* ML-35p, ATCC 25 922, and 54 080 [52]. Hb 98-114, a hemoglobin-derived AMP from the midgut of cattle tick (*Rhipicephalus microplus*), is the first hemoglobin reported with specific antifungal activity [52,53]. In addition, HbbetaP-1, derived from hemoglobin beta-subunit in Channel catfish (*Ictalurus punctatus*), was proven to have an anti-parasite activity against *Ichthyophthirius multifiliis* and *Tetrahymena pyriformis* [54]. In summary, hemoglobin’s activities against bacteria, fungal and parasites, suggesting their potential roles in enhancing immune defense. Therefore, the representative *Hemo2* may be useful for practical aquaculture of seahorses.

### 3.4. Sex Differences of Putative AMP/AMP Precursor Genes

In the present study, we identified three sex-differential putative AMP precursors based on both transcriptome and proteome data. Among them, two male-biased putative AMP precursors (*CDK-like16* and *CDK-like23*) were aligned to Scolopendin1 (query ID 2453 in Appendix A), which was originally identified from Chinese red-headed centipede (*Scolopendra subspinipe*) with antimicrobial and antifungal activities [23]. Other Scolopendins, like Scolopendin and Scolopendin2, was also proven to have broad-spectrum antimicrobial and antifungal activities [55,56,57]. Although the relationships between male pregnancy and *CDK-like* genes are still unknown, their antimicrobial activities support their potential applications in the improvement of aquaculture qualities of seahorses.

## 4. Materials and Methods

### 4.1. Data Collection

A total of 2,624 AMP amino-acid sequences (Appendix A) were downloaded from the public APD3 database. Genome sequence of the lined seahorse [26], previously reported by us, included assembled genome scaffolds and an annotated gene set. The 19 transcriptome raw datasets, covering multiple developmental stages and various tissues, were publically available in our another paper [1]. We applied a reference genome-based strategy to analyze the transcriptome data. Firstly, the transcriptome raw reads [1] were filtered by SOAPnuke (version 1.5.6) [58] to remove low-quality reads, and those reads with adapter sequences or PCR duplicates. Subsequently, clean reads were aligned to the assembled genome [26] with HISAT2 (version 2.0.4) [59]. Finally, we employed RSEM (version 1.2.12) [60] to calculate the FPKM value of each transcript. The proteome data were generated by us in this study (see more details in Section 4.3).

### 4.2. Prediction and Identification of Putative AMPs/AMP Precursors in the Lined Seahorse

We used BLAST to predict AMPs, with the 2624 AMP sequences as the queries and above-mentioned 21 multi-omics datasets as the subject sequences. Before running the BLAST (e-value = 1 × 10^−5^), we applied a makeblastdb command to build an index separately. We subsequently used in-house scripts to deal with alignment information, filtered those hits with query align ratios less than 0.5 and removed redundant data. The aligned region sequences and full-length gene sequences were captured from the archived subject sequences by an in-house script. Classification of putative AMPs was referred to as the annotation of AMPs in the AMD3 database. Finally, we obtained the predicted AMP/AMP precursor genes or sequences with detailed information for each hit.

### 4.3. Proteomics of the Lined Seahorse

#### 4.3.1. Identification by Mass Spectrometry (MS)

##### 4.3.1.1. Sample Preparation and Protein Extraction

We collected ten 5-month-old lined seahorses, with half male and half female, from BGI-Marine Aquaculture Base in Baguang, Shenzhen, Guangdong Province, China. Abdominal tissues for identification proteomics included skin and muscle, which were cut from the abdomen of each anesthetized seahorse. Tissues were vortexed with 5-mm magnetic beads, and lysis buffer (8 M urea, 40 mM Tris-HCl, 2 mM ethylenediaminetetraacetic acid (EDTA), 1 mM Phenylmethanesulfonyl fluoride (PMSF), pH 8.5). Subsequently, tissues were homogenized in a tissue lyser for two min. The samples were centrifuged at 25,000× *g* at 4 °C for 20 min. The supernatant liquid was reduced in 10 mM dithiothreitol (DTT; Amresco, Solon, OH, USA) at 56 °C for one hour prior to alkylation in 55 mM iodoacetamide (Sigma, St. Louis, MO, USA) at room temperature in the dark for 30 min. The protein concentration was determined by a Bradford kit (Bio-Rad, Hercules, CA, USA).

##### 4.3.1.2. Protein Digestion and Peptide Fractionation

Fractions of 100-μg alkylated proteins were diluted (to less than 2 M urea) with 50 mM tetraethylammonium borohydride (TEAB; Applied Biosystems, Milan, Italy), and then digested by trypsin at a ratio 1:40 (enzyme to protein) at 37 °C with overnight incubation. The digested peptides were desalted using a Strata X C18 column (Phenomenex, Torrance, CA, USA) according to the manufacturer’s instructions. The peptides then were dried by a speed-vac (LaboGene, Lynge, Denmark).

Peptides from three male (MS.M.20) and three female (MS.F.20) seahorses were performed with an extra procedure of fractionation. They were re-dissolved by 2-mL buffer A (5% acetonitrile (ACN), pH 9.8), and then separated by a Gemini C18 column (5 μm, 250 × 4.6 mm; Phenomenex) equipped on a Shimadzu LC-20AB system (Shimadzu Corporation, Kyoto, Japan). Peptides were fractionated using a gradient of buffer B (95% ACN, pH 9.8) for 10 min, 5–35% buffer B for 40 min, 35–95% buffer B for 1 min, buffer B for 3 min and 5% buffer B for 10 min at a constant flow rate of 1 mL/min. Fractions were collected each min and pooled into 20 fractions [61]. Subsequently, each fraction was dried by speed-vac (LaboGene).

##### 4.3.1.3. Identification of MS Acquisition

The dried fractionated peptides were re-dissolved in buffer C (2% ACN, 0.1% formic acid (FA)) and centrifuged at 20,000× *g* for 10 min to collect supernatants. Peptides were separated by Shimadzu LC-20AD, on which a 300 μm × 4 mm (μ-Precolumn, Thermo Scientific, Waltham, MA, USA) for peptide enrichment and a 75 μm × 15 cm in-house column for peptide separation were equipped. After enrichment and desalting in a trap column, each fraction was separated in in-house column by a 65-min gradient at 300 nl/min: 5% buffer D (98% ACN, 0.1% FA) for 8 min, 8–35% buffer D for 35 min, 35–60% buffer D for 5 min, 60–80% buffer D for 2 min, 80% buffer D for 5 min, and 5% buffer D for 10 min. Peptides were detected by the Q-Exactive tandem mass spectrometer (ThermoFisher Scientific, San Jose, CA, USA) that coupled to the Shimadzu LC-20AD at a data-dependent acquisition (DDA) mode. MS parameters were set as follows: spray voltage 1.6kv; scan range 350–1600 m/z; MS resolution 70,000; MS/MS resolution 17,500; dynamic exclusion duration 15s.

##### 4.3.1.4. MS Raw Data Analysis

MS raw data were converted to MGF files and searched against the annotated gene set of the lined seahorse with Mascot 2.3.02 [62]. Mascot parameters were listed as follows: enzyme: trypsin; peptide Mass tolerance: 20 ppm; fragment Mass tolerance: 0.05 Da; fixed modifications: Carbamidomethyl (C); variable modifications: Oxidation (M), Deamidation (N, Q); max missed cleavages: 1. To perform quality control for identification, peptide spectrum matches (PSMs) were re-scored by Percolator [63]. Subsequently, a protein inference was performed using Occam’s razor approach. The protein false discovery rate (FDR) was calculated by Pickedprotein FDR [64]. Finally, the credible protein identifications were obtained with FDR ≤ 0.01 at both PSM and protein levels.

#### 4.3.2. Quantification by Data Independent Acquisition (DIA)

To acquire the expression level of each identified protein, we performed DIA quantification of individual male and female seahorse. We fully grounded the five-month-old male and female seahorses and collected the homogenates. Protein extraction, digestion, and peptide fraction were performed as mentioned in Section 4.3.1.1 and Section 4.3.1.2 except that peptides were separated into 10 fractions in the DDA mode and without fractionation in the DIA mode.

The procedure of MS identification is similar to Section 4.3.1.3 but using different instruments and parameters. In brief, re-dissolved peptides were separated in a 150 μm × 25 cm in-house column equipped on UltiMate 3000 UHPLC (ThermoFisher Scientific) by a 180 min gradient at 500 nL/min: 5% buffer D (98% ACN, 0.1% FA) for 5 min, 5–35% buffer D for 155 min, 35–80% buffer D for 10 min, 80% buffer D for 5 min, and 5% buffer D for 5 min. Fractionated peptides were detected by the Q-Exactive HF (ThermoFisher Scientific) at the DDA and DIA modes, respectively. MS parameters were set as follows: spray voltage, 1.6kv; scan range, 350–1500 m/z; MS resolution at DDA mode, 60,000; MS resolution at DIA mode, 120,000; MS/MS resolution, 15,000; dynamic exclusion duration, 30s.

The DDA MS raw data were identified by MaxQuant (Version 1.5.3.30) [65], and the identification with FDR ≤ 0.01 on both PSM and protein levels was used to build a spectra library [65] for subsequent DIA analysis. MaxQuant parameters were set as follows: enzyme, trypsin; minimal peptide length, 7; PSM-level FDR, 0.01; protein-level FDR, 0.01; fixed modifications: Carbamidomethyl (C); variable modifications: Oxidation (M), Acetyl (Protein N-term); Database, the gene set of the lined seahorse. Based on the spectrum library of DDA data, DIA data was analyzed by Spectronaut 12 [66]. The retention time was corrected by the Spectronaut with indexed retention time (iRT). FDR was calculated by the mProphet approach in Spectronaut.

### 4.4. Validation of the Proteome Data

We integrated the proteome data and removed those repeat sequences to obtain the final non-redundant protein lists of the total, female, and male lined seahorses. These identified protein lists were mapped to the predicted AMP/AMP precursor genes for overlapping determination.

## 5. Conclusions

Using a high-throughput strategy, we identified 290 putative AMPs/AMP precursors from our previously reported genome and transcriptome data of the lined seahorse. Among them, 267 are novel, and 118 were further validated by the proteome data that were generated in this study. Transcription levels of top 20 putative AMPs/AMP precursors (with the highest FPKM values) revealed tissue preference of different AMP groups, especially for lectins. For example, *Lectin25* was highly transcribed at the pre-pregnancy and pregnancy stages in the pouch. These data are consistent with previous reports that Lectin25 is involved in seahorse male pregnancy. A comparison between one-day and three-month juveniles indicated that *Hemo2* was significantly upregulated in the seahorse body, haslet, and the brain, suggesting that Hemo2 may be useful for the treatment of the popular lethal enteritis in juveniles. Proteome validation of the female and male seahorses identified three sex-differential AMP precursors. Among them, CDK-like16 and CDK-like23 were male-biased. In summary, we identified putative AMPs and AMP precursors from multi-omics data of the lined seahorse. These data will contribute to the development of AMP-based fish food additives and human drugs.

## Figures and Tables

**Figure 1 marinedrugs-18-00030-f001:**
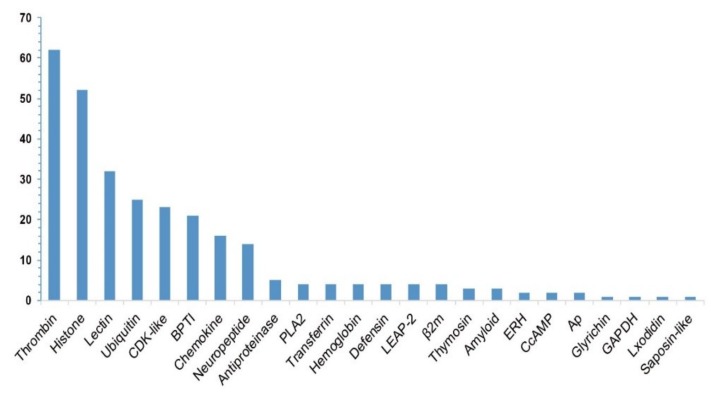
Summary of the identified putative AMP/AMP precursor numbers (the y-axis) from the lined seahorse. A total of 290 putative AMP sequences were determined with a subsequent categorization of 24 groups (the x-axis represents each classified group).

**Figure 2 marinedrugs-18-00030-f002:**
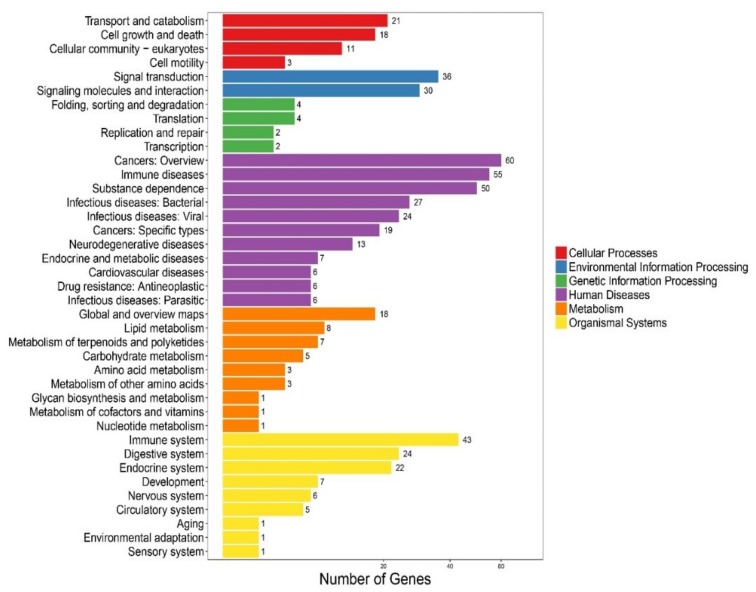
KEGG annotation of the putative AMP/AMP precursor genes in the lined seahorse. A total of 272 predicted genes were clustered into 193 KEGG pathways.

**Figure 3 marinedrugs-18-00030-f003:**
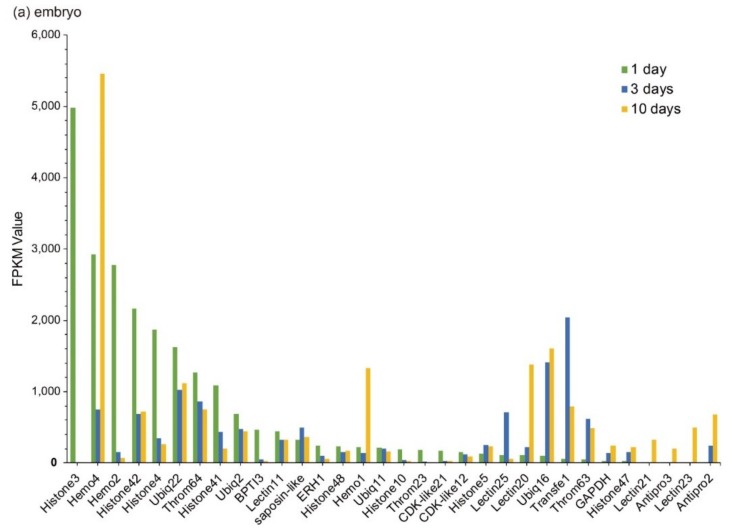
Top 20 putative AMPs/AMP precursors based on their transcription levels (FPKM values) at different stages. Various tissues include (**a**) embryo, (**b**) brain, (**c**) testis, and (**d**) pouch. The x-axis is the identified putative AMPs/AMP precursors, and the y-axis represents the FPKM values.

**Figure 4 marinedrugs-18-00030-f004:**
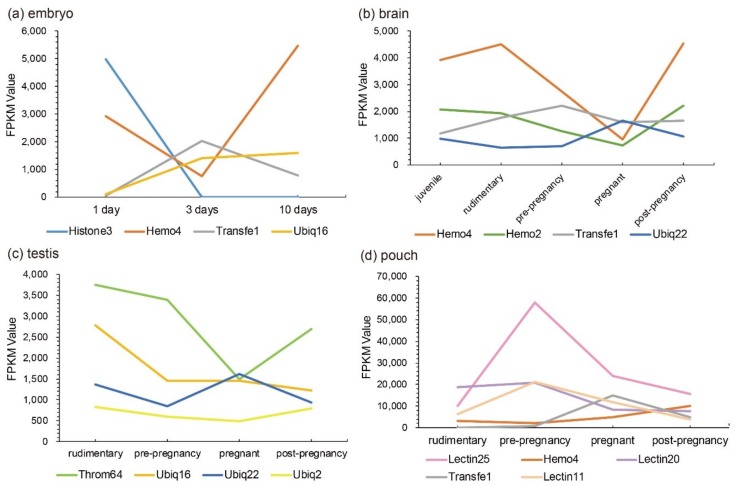
Representative AMPs or AMP precursors with significant changes (over two folds) or high transcription levels (FPKM values). Various tissues include (**a**) embryo, (**b**) brain, (**c**) testis, and (**d**) pouch. The x-axis is different stages (1 to 10-day embryos or from juvenile to post-pregnancy), and the y-axis represents FPKM values.

**Figure 5 marinedrugs-18-00030-f005:**
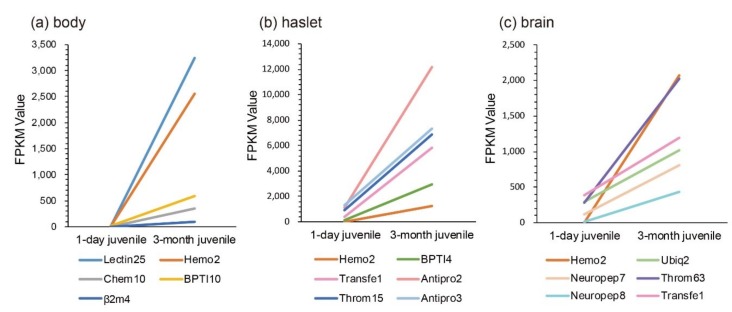
Transcriptional changes of representative AMPs or AMP precursors with significant upregulation and high FPKM values during the development from one-day to three-month juveniles. (**a**) body; (**b**) haslet; (**c**) brain. The *x*-axis is different stages, and the y-axis represents the FPKM values.

**Figure 6 marinedrugs-18-00030-f006:**
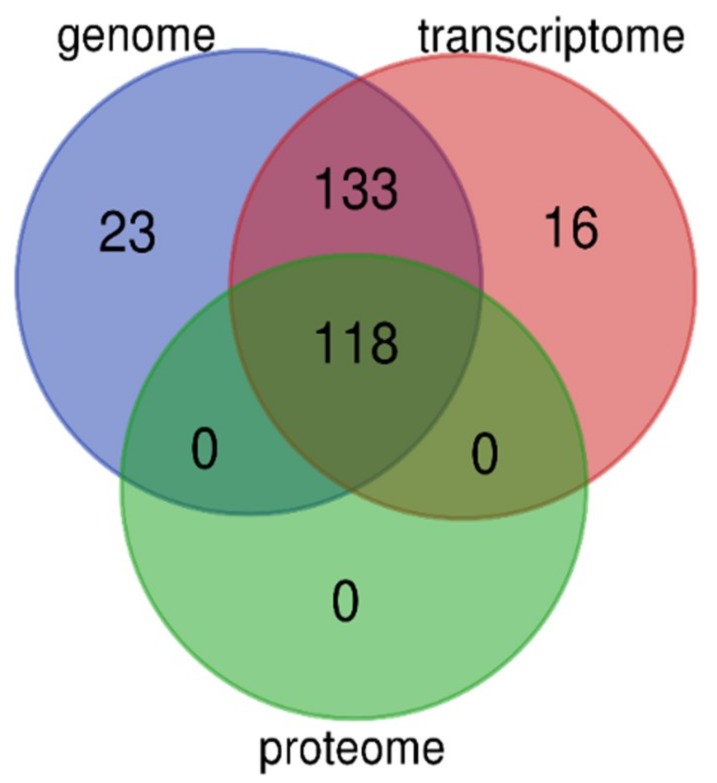
Overlapping of putative AMPs and AMP precursors identified from the genome, transcriptome, and proteome datasets. A total of 118 AMP sequences were validated by these multi-omics resources.

**Figure 7 marinedrugs-18-00030-f007:**
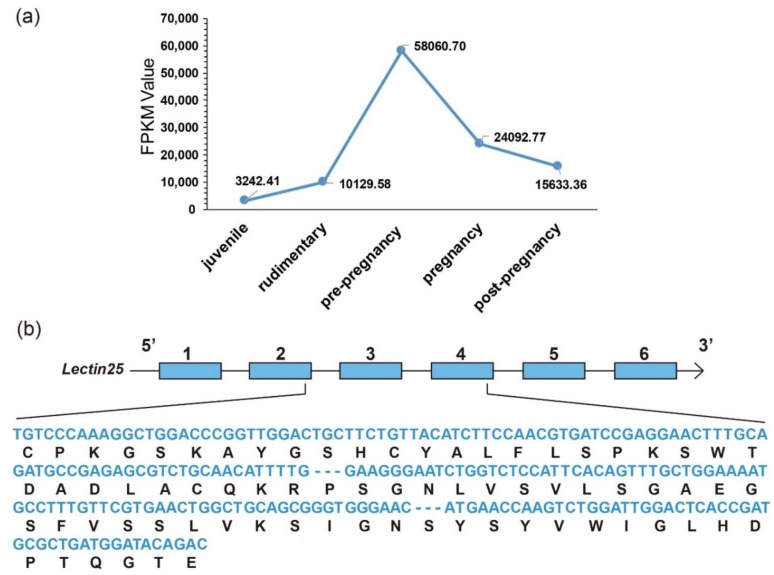
Representative *Lectin25* (Galactose-specific lectin nattectin). (**a**) Transcriptional changes (FPKM values, the y-axis) during development (the x-axis for various development stages) in the pouch. (**b**) Predicted gene structure with the aligned AMP sequences (nucleotides in blue, and translated amino-acids in black).

**Figure 8 marinedrugs-18-00030-f008:**
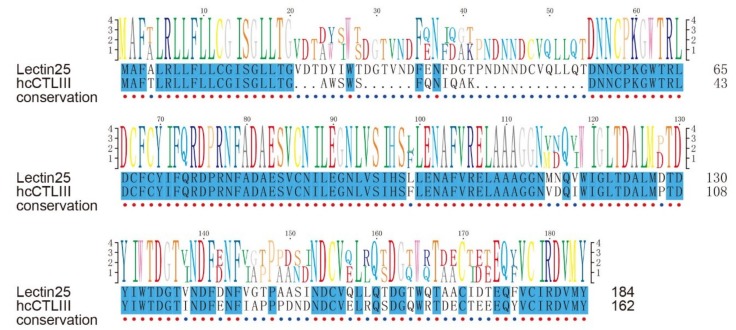
A Pairwise sequence alignment of Lectin25 and *hc*CTL III. Blue blocks mark the conservation (identity of 100%).

**Table 1 marinedrugs-18-00030-t001:** Statistics of the identified putative AMP/AMP precursor numbers from the transcriptome datasets [1] of various tissues.

Tissues	Juvenile ^1^	Rudimentary ^2^	Pre-Pregnancy ^3^	Pregnancy ^4^	Post-Pregnancy ^5^
Brain	233	234	237	240	241
testis	241 (haslet)	252	246	256	241
pouch	240 (body)	225	225	242	227

^1^ Three months post-birth; ^2^ Five months post-birth; ^3^ Seven months post-birth; ^4^ Approximately eight months post-birth, carrying 10-day embryos in the brood pouch; ^5^ Males collected in 10 days after being released from their offspring.

**Table 2 marinedrugs-18-00030-t002:** Statistics of the identified putative AMP/AMP precursor numbers from the transcriptome datasets [1] of embryos and juveniles.

Sample	1-D Embryo *	3-D Embryo *	10-D Embryo *	1-D Juvenile ^#^
Number	219	245	242	246

* 1/3/10-day post-fertilization; ^#^ Young seahorses collected entirely in the first day after hatching.

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
