# Peer review of "High-Throughput Identification of Putative Antimicrobial Peptides from Multi-Omics Data of the Lined Seahorse (Hippocampus erectus)"

_marinedrugs, 2019, doi:10.3390/md18010030_

Round 1

Reviewer 1 Report

Fig 1. Authors should represent amino acid sequence alignment with known one for each group of AMPs, as claimed as a novel. It is important.
Fig 3. What FPKM values in figure 3. Please provide the details in the legend at least once.
FPKM values are differing for different peptides in different tissues. What is the significance? Please discuss.
What is Y-axis in Fig 4, please describe in the figure legend.
In Fig 4. Transcriptional level showed for various tissues while in Fig 5. Showed for days 1 to 3 for different peptides. What is the significance? Please discuss why different peptides have involved in the analysis of transcriptional levels in tissues in comparison to days 1 to 3? Why the authors used different tissues to analyze the transcriptional level of peptides in these two Figures.
What does Fig 6, signifies?
Fig 7b. Please indicate the direction in gene structure and please label (functional annotation) the genes too.
Fig 8. Why authors used only one peptide for sequence alignment while claiming more 200 as a novel? Please provide the data for others too.

Author Response

Fig 1. Authors should represent amino acid sequence alignment with known one for each group of AMPs, as claimed as a novel. It is important.

Answer: Thanks for your advice. Yes, we marked the aligned regions for our identified 290 putative AMPs/AMP precursors (see more details in the new File S5) in the Supplementary Materials. In fact, it is difficult to display all these alignments directly. We hence marked only the aligned regions between the queries and our identified AMPs (with the yellow marks in the File S5). Here, we regard hits to queries with identity of 100% as known AMPs, while those with one or more residue change(s) as novel AMPs, which were described on lines 122-123 and marked in the File S2.

Fig 3. What FPKM values in figure 3. Please provide the details in the legend at least once.

FPKM values are differing for different peptides in different tissues. What is the significance? Please discuss. What is Y-axis in Fig 4, please describe in the figure legend.

Answer: Thanks for your questions and nice advice. In fact, the explanation of FPKM values (i.e. transcription levels; on line 209 of the figure legend of Fig 3) was provided in the main text under section 2.2 (on lines 155-156). We revised legends of both figures 3 and 4 on lines 208-210 and 211-214, respectively.

We don’t have biological repeats for the transcriptome datasets, since each reported transcriptome was sequenced from a single lined seahorse (lines 94-95); hence, we applied fold changes to represent significances (see more details on lines 172-173 and 198-199).

By the way, for the y-axis in Fig 4, we added its description on line 214.

In Fig 4. Transcriptional level showed for various tissues while in Fig 5. Showed for days 1 to 3 for different peptides. What is the significance? Please discuss why different peptides have involved in the analysis of transcriptional levels in tissues in comparison to days 1 to 3? Why the authors used different tissues to analyze the transcriptional level of peptides in these two Figures.

Answer: Thanks for your questions. We analyzed significantly up-regulated putative AMPs between 1-day and 3-month juveniles. As mentioned in the previous answer, we applied fold changes to represent significances (lines 208-210 and 211-214) in the revised figure legend (on line 211).

According to previous studies, juvenile seahorses within about one month after birth are more likely to be susceptible to exogenous bacterial infection, and enteritis occurs frequently in this period. However, we don’t have any transcriptome data for 1-month stage. Hence, we collected transcriptome data at the 3-month stage as an alternative. We aimed to screen those significantly up-regulated AMPs/AMP precursors that may contribute to seahorse immunity improvement (see more details on lines 198-200, 219-220, 230-232 and 343-344).

We used different tissues to analyze transcriptional changes of AMPs/AMP precursors in both Figures 4 and 5, because we have already discovered tissue preference of AMP distribution and the fact that expression of AMPs/AMP precursors varies among different tissues. Hence, we hope to explore more information in each tissue at the representative stages. Please find more details on lines 198-232 and 309-344.

What does Fig 6, signifies?

Answer: Thanks for your question. In fact, Figure 6 displayed overlapping of putative AMPs/AMP precursors among the genome, transcriptome and proteome datasets. It seems that among the 290 putative AMPs/AMP precursors, 251 were identified from both the genome and transcriptome data, and 118 were validated by the proteome data. Interestingly, however, 23 and 16 AMP/AMP precursors were not overlapped between the genome and transcriptome results. It seems that we can explore more AMPs by using different omics datasets, and the overlapping confirms existence of these identified AMP sequences from various resources. Please find corresponding discussions in the section 3.1 (lines 262-277 and 283-285).

Fig 7b. Please indicate the direction in gene structure and please label (functional annotation) the genes too.

Answer: Thanks for your advice. Yes, it is done in the revised Fig 7b (with 5’ to 3’ and an arrow for the gene direction) on page 10.

Fig 8. Why authors used only one peptide for sequence alignment while claiming more 200 as a novel? Please provide the data for others too.

Answer: Thanks for your question. Figure 8 displayed amino acid sequence alignment for the representative Lectin25 and hcCTL III to check their similarity. Interestingly, we found that hcCTL III mapped to Lectin25 with an identity of 80.2%. Hence, we speculate that Lectin25 may also play a similar role in embryo immune protection during male pregnancy.

As you noted, we collected 2,927 AMP amino acid sequences from the APD3 database (http://aps.unmc.edu/AP/main.php) as the query sequences, and aligned them to our genome and transcriptome data. Finally, we identified 290 putative AMPs/AMP precursors, of which 267 are novel with at least one-residue difference from their corresponding query sequences. Files S2, S3 and new S8 in the Supplementary Materials provide detailed alignment information of all the identified putative AMPs for readers’ reference. The explanation of novel was provided on lines of 122-123 of the revised manuscript.

By the way, we obtained editing help from a colleague, who had worked in the USA for over nine years.

Reviewer 2 Report

Reviewer's comments:

The manuscript from Xiyang Chen et al, report the identification of 290 putative AMPs from multi-omics (including genomic, transcriptomic and proteomic) data of the lined seahorse. Among these 290 putative AMPs, the authors claimed to have identified 267 novel AMPs and they have confirmed by proteomics that 118 of them are indeed produced in the lined seahorse. They use transcriptomic data to assess the regulation of a set of chosen AMPs in different tissues along the different stages of development. They also compared the AMPs specificity between males and females.

The work has been well performed, the methodology used is well appropriate for the objectives proposed, the results are in accordance with the technics used, there is a good coverage of the literature and the manuscript is clearly written.

However, there is a major confusion between “putative AMPs” and “putative AMPs precursors” in the manuscript that needs to be addressed. Therefore, some conclusions are drawn without sufficient proofs. The role of the “putative AMPs precursor” in the immune response should be nuanced and the authors should emphasize the “putative AMPs”, if they found any which isn’t very clear in the manuscript.

To conclude, regarding the Marine drugs journal notoriety, I believe that this article could be accepted after major revisions.

The authors should address the following questions.

Major Revisions

According to the literature and as it is mentioned in the introduction, some AMPs come from the hydrolysis of larger proteins such as histones, lectin, thrombin… Therefore, when the authors say they identified transcripts corresponding to these AMPs they are actually referring to the proteins that have the potential of being AMPs precursors after proteolytic cleavage. In my opinion, the genes or transcripts corresponding to these proteins should be then referred as “putative AMPs precursors” instead of “putative AMPs”. Moreover, the proteomics studies can’t discriminate the presence of the AMPs or their precursors as the samples for these studies are treated with trypsin. Moreover, there is no proof that the AMPs coming from bigger proteins are indeed produced and maintained in the lined seahorse. For example, Buforin I (which is cited in the introduction of this paper) is an AMP derivate from Histone H2A by pepsin-directed hydrolysis. Consequently, detection of transcript of Histone H2A in the brain for example can’t be correlated with production of Buforin I as it requires an enzyme that is specific of the gastric compartment. The same reasoning applies to other precursors of AMPs, their presence doesn’t necessary mean that the AMPs are produced as well. The distinction between “AMPs” and “AMPs precursors” should be corrected throughout the entire paper.

Therefore, in my opinion, the omics-studies of the “AMPs precursors” can’t be used to draw conclusions on the immune adaptive response of the lined seahorse. If the omics-studies were done comparing healthy versus infected animals and variation expression of bigger proteins that are putative AMPs precursors would be observed, then the authors could suggest a role of these proteins in the immune system. However, in the study presented, the only conclusions that can be drawn are about the role of these proteins in the physiological development of the lined seahorse. AMPs levels could vary during development because of the higher/lower production of precursor proteins however this is dependent of the proteolysis regulation of these proteins and nothing is shown or discussed on that topic in this manuscript.

Page 1 – line 4 and everywhere: The authors used the ‘Hippocamous erectus’ as Latin name instead of ‘Hippocampus erectus’. Thus, I suggest to use the right term all over the manuscript.

Page 5 - line 185 to 190: The authors wrote “Representative AMPs with high transcription levels or significant changes in each tissue were picked out. Their change patterns at different developmental stages were marked in Figure 4. These various patterns support that AMPs work together to provide differential protection against exogenous microbial.” I don’t think the authors can draw this conclusion for the transcriptomic analysis. As most of the “AMPs” identified actually correspond to bigger proteins that could be AMP precursors but that also have functions others than antimicrobial, aren’t these patterns just representative of the role of these proteins during development?

Page 9 – line 212: The authors wrote “Hemo2 was upregulated significantly in the examined tissues, suggesting its crucial role in development of the immune system. Our Swiss-Prot annotation indicated that Hemo2 was predicted to be hemoglobin subunit alpha-1.” The authors previously specified on page 5: “Synthesized peptide of hemoglobin β1, also identified by the present high-throughput method, has been proven to have high inhibitive activities on Micrococcus luteus”. However again there is no proof that hemoglobin subunit alpha-1 exists by itself in the lined seahorse and not only in the hemoglobin complex.

Page 10 – line 251 to 252: “Especially, the proteome data provide strong evidence for verification of these AMPs.” The proteome data provide strong evidence of the protein production. Because of the trypsin digestion to analyze the data, it is not possible to know if the peptides are produced in the lined seahorse or if they come from the trypsin digestion of potential AMP precursors.

Minor Revisions

Abstract

Page 1 – line 17-19: The first sentence is confusing. Are the authors saying that the demand for lined seahorses is high compared to the offer because of its high medicinal value? Then “for” on line 18 should probably be replaced by “because of”. Or are the supplies for medical applications low compared to other uses of the lined seahorse?

Introduction

Page 2 – line 54: The authors wrote “Enteritis is a serious disease to restrict the practical aquaculture…” it seems the appropriate phrasing is “Enteritis is a serious disease restricting the practical…”

Page 2 – line 57: It could be appropriate to include a reference on drug resistance emerging from the aquaculture. There is quite a number of review on that matter.

Page 2 – line 84: The authors wrote: “The existence of potential AMPs would be validated at the protein level based on our newly generated proteome data in the present study.” It seems more appropriate to write: “The production in the lined seahorse of some potential AMPs is validated… in the present study.”

Results

Page 2 – line 90: The authors wrote “We collected 2,927 AMP protein sequences”, the word “protein” should be removed and “AMP” should be replaced by “AMP or AMP precursor”.

Page 4 – line 134-136: “Representative classes include cancers: overview, immune diseases, substance dependence, and immune system (see Figure 2), indicating that most of the putative AMPs are potentially involved in immune and disease protection”. The “:” after “cancers” make the sentence confusing. If the authors wants to use the exact labels of the category they should probably use quote marks for each category.

As the paper suggests that the newly discovered AMPs could be useful to replace antibiotics in aquaculture to fight enteritis, it would be relevant to mention the “infectious diseases” categories which taken together regroup more AMP genes than the “immune diseases” or “substance dependence” for example.

There is also a significant amount of genes in the environmental information processing category, it could be interesting to discuss them.

Page 6 to 7 – Figure 3: The color code is confusing. The colors are maintained throughout Figure 3c and 3d but not 3b.

Page 8 - Figure 4: Same commentary, for more clarity the color code should be maintained throughout the figure (1 protein = 1 color).

Page 8 – Figure 5: Idem

Page 9 – line 212: However is not well suited here, it could be replaced by “Interestingly, Notably…”. I guess the authors meant “upregulated significantly in all examined tissues…”.

Author Response

The manuscript from Xiyang Chen et al, report the identification of 290 putative AMPs from multi-omics (including genomic, transcriptomic and proteomic) data of the lined seahorse. Among these 290 putative AMPs, the authors claimed to have identified 267 novel AMPs and they have confirmed by proteomics that 118 of them are indeed produced in the lined seahorse. They use transcriptomic data to assess the regulation of a set of chosen AMPs in different tissues along the different stages of development. They also compared the AMPs specificity between males and females.

The work has been well performed, the methodology used is well appropriate for the objectives proposed, the results are in accordance with the technics used, there is a good coverage of the literature and the manuscript is clearly written.

However, there is a major confusion between “putative AMPs” and “putative AMPs precursors” in the manuscript that needs to be addressed. Therefore, some conclusions are drawn without sufficient proofs. The role of the “putative AMPs precursor” in the immune response should be nuanced and the authors should emphasize the “putative AMPs”, if they found any which isn’t very clear in the manuscript.

To conclude, regarding the Marine drugs journal notoriety, I believe that this article could be accepted after major revisions.

Answer: Thanks for your positive comments and nice advice. Yes, you are right. When necessary, we corrected AMPs as “putative AMPs and AMP precursors” throughout the revised manuscript. Potential immune roles of the putative AMP precursors were not emphasized.

By the way, we obtained editing help from a colleague, who had worked in the USA for over nine years.

The authors should address the following questions.

Major Revisions

According to the literature and as it is mentioned in the introduction, some AMPs come from the hydrolysis of larger proteins such as histones, lectin, thrombin… Therefore, when the authors say they identified transcripts corresponding to these AMPs they are actually referring to the proteins that have the potential of being AMPs precursors after proteolytic cleavage. In my opinion, the genes or transcripts corresponding to these proteins should be then referred as “putative AMPs precursors” instead of “putative AMPs”.

Answer: Thanks for your advice. Yes, the term was revised throughout our revised manuscript in accordance with your instructions.

Moreover, the proteomics studies can’t discriminate the presence of the AMPs or their precursors as the samples for these studies are treated with trypsin.

Answer: Thanks for your comments. Yes, you are right theoretically. However, we could detect AMP regions with extended sequences at either side or both sides for our precursors. That is to say, we can tell the difference between free AMPs and proteins-derived AMPs. Please find more details in the new File S5.

Moreover, there is no proof that the AMPs coming from bigger proteins are indeed produced and maintained in the lined seahorse. For example, Buforin I (which is cited in the introduction of this paper) is an AMP derivate from Histone H2A by pepsin-directed hydrolysis. Consequently, detection of transcript of Histone H2A in the brain for example can’t be correlated with production of Buforin I as it requires an enzyme that is specific of the gastric compartment. The same reasoning applies to other precursors of AMPs, their presence doesn’t necessary mean that the AMPs are produced as well. The distinction between “AMPs” and “AMPs precursors” should be corrected throughout the entire paper.

Answer: Thanks for your nice advice and comments. Yes, as you stated, we corrected these terms as “putative AMPs and AMP precursors” throughout the revised manuscript.

Therefore, in my opinion, the omics-studies of the “AMPs precursors” can’t be used to draw conclusions on the immune adaptive response of the lined seahorse. If the omics-studies were done comparing healthy versus infected animals and variation expression of bigger proteins that are putative AMPs precursors would be observed, then the authors could suggest a role of these proteins in the immune system. However, in the study presented, the only conclusions that can be drawn are about the role of these proteins in the physiological development of the lined seahorse. AMPs levels could vary during development because of the higher/lower production of precursor proteins however this is dependent of the proteolysis regulation of these proteins and nothing is shown or discussed on that topic in this manuscript.

Answer: Sorry for the misleading descriptions. Yes, you are right. We can’t draw conclusions on the immune adaptive responses of the AMP precursors, although some had been proven to be antibacterial. We corrected related statements all over the revised manuscript.

We didn’t compare changes of putative AMPs or AMP precursors between healthy and infected lined seahorses in the present work, so we can’t illustrate the roles of these putative AMP precursors in the immune system. However, transcription levels of putative AMP precursors during development may provide instructive clues to discover significant roles at particular stages or in special tissues. Anyway, we carefully revised our related conclusions and added a paragraph to discuss this problem on lines 283-290.

Page 1 – line 4 and everywhere: The authors used the ‘Hippocamous erectus’ as Latin name instead of ‘Hippocampus erectus’. Thus, I suggest to use the right term all over the manuscript.

Answer: Sorry for this mistake, which was corrected throughout our revised manuscript.

Page 5 - line 185 to 190: The authors wrote “Representative AMPs with high transcription levels or significant changes in each tissue were picked out. Their change patterns at different developmental stages were marked in Figure 4. These various patterns support that AMPs work together to provide differential protection against exogenous microbial.” I don’t think the authors can draw this conclusion for the transcriptomic analysis. As most of the “AMPs” identified actually correspond to bigger proteins that could be AMP precursors but that also have functions others than antimicrobial, aren’t these patterns just representative of the role of these proteins during development?

Answer: Sorry for the misleading descriptions. You are right. Our conclusions based on transcriptome data are not solid enough. Hence, we delete this sentence and modified the statement in accordance with your comment as follows (on lines 198-203).

Representative putative AMPs or AMP precursors with high transcription levels or significant changes (at least two folds) in each tissue were picked out. Their change patterns at different developmental stages were marked in Figure 4. We speculate that those putative AMPs and AMP precursors with high transcription levels or significant fold changes at certain stage(s) may play certain vital roles, which are worthy of in-depth investigations into their disease resistance for aquaculture practices.

Page 9 – line 212: The authors wrote “Hemo2 was upregulated significantly in the examined tissues, suggesting its crucial role in development of the immune system. Our Swiss-Prot annotation indicated that Hemo2 was predicted to be hemoglobin subunit alpha-1.” The authors previously specified on page 5: “Synthesized peptide of hemoglobin β1, also identified by the present high-throughput method, has been proven to have high inhibitive activities on Micrococcus luteus”. However again there is no proof that hemoglobin subunit alpha-1 exists by itself in the lined seahorse and not only in the hemoglobin complex.

Answer: Thanks for your nice comments. Yes, we rephrased the sentence on lines 176-183 and deleted the sentence of “Our Swiss-Prot annotation indicated that Hemo2 was predicted to be hemoglobin subunit alpha-1” on line 227.

Page 10 – line 251 to 252: “Especially, the proteome data provide strong evidence for verification of these AMPs.” The proteome data provide strong evidence of the protein production. Because of the trypsin digestion to analyze the data, it is not possible to know if the peptides are produced in the lined seahorse or if they come from the trypsin digestion of potential AMP precursors.

Answer: Thanks for your question. Yes, you are right. In fact, we performed validation of AMP peptide sequences. In addition, the proteome data only provide evidence of the precursor production rather than AMP production, so “AMPs” was replaced by “AMP precursors” throughout the vised manuscript.

Minor Revisions

Abstract

Page 1 – line 17-19: The first sentence is confusing. Are the authors saying that the demand for lined seahorses is high compared to the offer because of its high medicinal value? Then “for” on line 18 should probably be replaced by “because of”. Or are the supplies for medical applications low compared to other uses of the lined seahorse?

Answer: Thanks for your comments. You are right. It was corrected on line 18, changing “for ” with “because of”.

Introduction

Page 2 – line 54: The authors wrote “Enteritis is a serious disease to restrict the practical aquaculture…” it seems the appropriate phrasing is “Enteritis is a serious disease restricting the practical…”

Answer: Sorry for the misleading description. Yes, it was corrected on line 54.

Page 2 – line 57: It could be appropriate to include a reference on drug resistance emerging from the aquaculture. There is quite a number of review on that matter.

Answer: Thanks for your nice advice. We cited a review paper [14] on line 57.

Page 2 – line 84: The authors wrote: “The existence of potential AMPs would be validated at the protein level based on our newly generated proteome data in the present study.” It seems more appropriate to write: “The production in the lined seahorse of some potential AMPs is validated… in the present study.”

Answer: Thanks for your nice advice. Yes, this sentence was corrected on lines 85-86.

Results

Page 2 – line 90: The authors wrote “We collected 2,927 AMP protein sequences”, the word “protein” should be removed and “AMP” should be replaced by “AMP or AMP precursor”.

Answer: Thanks for your nice advice. Yes, the word “protein” was replaced by “amino acid” on line 91. We also corrected AMP as “AMP or AMP precursor” throughout the revised manuscript

Page 4 – line 134-136: “Representative classes include cancers: overview, immune diseases, substance dependence, and immune system (see Figure 2), indicating that most of the putative AMPs are potentially involved in immune and disease protection”. The “:” after “cancers” make the sentence confusing. If the authors wants to use the exact labels of the category they should probably use quote marks for each category.

Answer: Sorry for this misleading writing. As you recommended, we applied a quote mark for each category on lines 144.

As the paper suggests that the newly discovered AMPs could be useful to replace antibiotics in aquaculture to fight enteritis, it would be relevant to mention the “infectious diseases” categories which taken together regroup more AMP genes than the “immune diseases” or “substance dependence” for example. There is also a significant amount of genes in the environmental information processing category, it could be interesting to discuss them.

Answer: Thanks for your advice and comments. Yes, we mentioned it on lines 53-54 and 143-148. As to those genes clustered into the environmental information processing category, however, we haven’t observed something interesting to discuss yet.

Page 6 to 7 – Figure 3: The color code is confusing. The colors are maintained throughout Figure 3c and 3d but not 3b.

Answer: Sorry for this mistake. Yes, we changed the colors with codes in the revised Figure 3b so as to keep consistence with those in Fig 3c and 3d. Please find these corrections on page 6.

Page 8 - Figure 4: Same commentary, for more clarity the color code should be maintained throughout the figure (1 protein = 1 color). Page 8 – Figure 5: Idem

Answer: Thanks for your advice. Yes, we changed colors in the revised Figures 4 and 5 (on page 8).

Page 9 – line 212: However is not well suited here, it could be replaced by “Interestingly, Notably…”. I guess the authors meant “upregulated significantly in all examined tissues…”.

Answer: Yes, we corrected it with “Notably” on line 227.

Round 2

Reviewer 1 Report

Authors did a satisfactory response to all the queries and suggestions. I would accept the manuscript in the present form.